# S100A4 Is a Strong Negative Prognostic Marker and Potential Therapeutic Target in Adenocarcinoma of the Stomach and Esophagus

**DOI:** 10.3390/cells11061056

**Published:** 2022-03-21

**Authors:** Christoph Treese, Kimberly Hartl, Michelle Pötzsch, Matthias Dahlmann, Moritz von Winterfeld, Erika Berg, Michael Hummel, Lena Timm, Beate Rau, Wolfgang Walther, Severin Daum, Dennis Kobelt, Ulrike Stein

**Affiliations:** 1Experimental and Clinical Research Center, Charité-Universitätsmedizin and Max-Delbrück-Center for Molecular Medicine in the Helmholtz Association, 10115 Berlin, Germany; christoph.treese@charite.de (C.T.); or kimberly.hartl@charite.de (K.H.); dahlmann@mdc-berlin.de (M.D.); wowalt@mdc-berlin.de (W.W.); Dennis.Kobelt@epo-berlin.com (D.K.); 2Department of Gastroenterology, Infectious Diseases and Rheumatology, Campus Benjamin Franklin, Charité-Universitätsmedizin Berlin, Corporate Member of Freie Universität Berlin and Humboldt-Universität zu Berlin, 10115 Berlin, Germany; poetzsch.michelle@gmail.com (M.P.); lena.timm@mdc-berlin.de (L.T.); severin.daum@charite.de (S.D.); 3Berlin Institute of Health (BIH), 10115 Berlin, Germany; 4Medical Department, Division of Gastroenterology and Hepatology, Campus Virchow-Klinikum, Charité-Universitätsmedizin Berlin, Corporate Member of Freie Universität Berlin and Humboldt-Universität zu Berlin, 10115 Berlin, Germany; 5Berlin Institute for Medical Systems Biology, Max Delbrück Center for Molecular Medicine, 10115 Berlin, Germany; 6Institute of Pathology, Charité-Universitätsmedizin Berlin, Corporate Member of Freie Universität Berlin and Humboldt-Universität zu Berlin, 10115 Berlin, Germany; m.vonwinterfeld@pathologie-rosenheim.de (M.v.W.); erika.berg@charite.de (E.B.); michael.hummel@charite.de (M.H.); 7Department of Surgery, Campus Virchow-Klinikum and Campus Mitte, Charité-Universitätsmedizin Berlin, Corporate Member of Freie Universität Berlin and Humboldt-Universität zu Berlin, 10115 Berlin, Germany; beate.rau@charite.de; 8German Cancer Consortium (DKTK), 69126 Heidelberg, Germany

**Keywords:** gastric cancer, esophageal cancer, metastasis, S100A4, niclosamide

## Abstract

Deregulated Wnt-signaling is a key mechanism driving metastasis in adenocarcinoma of the gastroesophageal junction and stomach (AGE/S). The oncogene *S100A4* was identified as a Wnt-signaling target gene and is known to promote metastasis. In this project, we illuminate the role of S100A4 for metastases development and disease prognosis of AGE/S. Five gastric cancer cell lines were assessed for S100A4 expression. Two cell lines with endogenous high S100A4 expression were used for functional phenotyping including analysis of proliferation and migration after stable *S100A4* knock-down. The prognostic value of S100A4 was evaluated by analyzing the S100A4 expression of tissue microarrays with samples of 277 patients with AGE/S. *S100A4* knock-down induced lower migration in FLO1 and NCI-N87 cells. Treatment with niclosamide in these cells led to partial inhibition of S100A4 and to reduced migration. Patients with high S100A4 expression showed lower 5-year overall and disease-specific survival. In addition, a larger share of patients in the S100A4 high expressing group suffered from metachronous metastasis. This study identifies S100A4 as a negative prognostic marker for patients with AGE/S. The strong correlation between S100A4 expression, metastases development and patient survival might open opportunities to use S100A4 to improve the prognosis of these patients and as a therapeutic target for intervention in this tumor entity.

## 1. Introduction

In 2020, approximately 769,000 deaths were caused by esophageal and gastric adenocarcinomas (AGE/S) worldwide [1]. The prognosis for these tumor entities is comparatively poor with an average 5-year patient survival of less than 30% [2,3]. A major problem treating AGE/S is the asymptomatic development over a longer period. At the time of diagnosis, lymph node metastases are detected in 50% and distant metastases in 30% of patients [1,4,5]. Another challenge is that microscopic, radiologically undetectable metastasis also occurs in the early stages of the disease and thus leads to misjudged tumor staging and decision-making in therapy [6].

To improve the prognosis of AGE/S patients, biomarkers are needed that enable the detection of primary diseases with high metastatic potential in any tumor stage. Besides this prognostic aspect, also new, targeted therapy approaches are necessary to improve patient survival in a metastatic situation.

The calcium-binding protein S100A4 was first described as a metastasis-inducing protein in breast cancer in 1993 [7]. Further research revealed that this target gene of β-catenin/TCF plays a role in metastases formation, by inducing for instance migration, invasion, adhesion, and angiogenesis [8]. S100A4′s metastasis inducing potential is not only important in breast cancer but in many other solid tumor entities such as lung, colon, gastric, esophageal, hepatocellular, pancreatic, gallbladder, ovarian and prostate cancer [9].

Yet, S100A4 expression in gastric cancer has been analyzed exclusively in studies with patient cohorts of Asian origin. These studies revealed S100A4 expression in tumor tissue as a negative prognostic marker and associated with the presence of distant metastasis [10,11,12,13,14,15]. Interestingly a meta-analysis of seven gastric cancer studies did not recapitulate that correlation [16].

Currently, there is only one study focusing on *S100A4* status in AGE/S in a Caucasian cohort published. This study analyses *S100A4* transcripts in the blood plasma and finds a positive correlation between *S100A4* detection, poor survival and metastasis formation [17]. However, no expression analysis of tumor tissue was performed.

Besides its high potential as a prognostic marker, S100A4 is also an interesting therapeutic target. Former high throughput screenings have identified potent *S100A4* inhibitors such as calcimycin, niclosamide or sulindac that decreased motility and invasiveness of colon cancer cells in vitro and metastasis development in vivo [18,19,20]. Niclosamide represses transcription of *S100A4* by inhibiting β-catenin expression and its binding to TCF (T-cell factor) which results in reduced migration of cells and lower metastasis burden in mice with colorectal carcinoma [18,21]. It is approved by FDA (Food and Drug Administration in the USA) and EMA (European Medicines Agency in the EU) for other indications. A phase II clinical trial analyzing the safety and efficacy of oral niclosamide application in colorectal patients with distant metastasis is already ongoing (NCT02519582) [22]. This pharmaceutical could be easily repositioned for use in AGE/S.

In this study, we analyzed the potential of S100A4 expression as a prognostic marker in AGE/S and examined both the effect of *S100A4* knock-down and the effect of a pharmacological *S100A4* suppression in AGE/S cell lines.

## 2. Materials and Methods

### 2.1. Patients and TMA

The study cohort consists of 277 patients with AGE/S of all tumor stages, primarily treated by surgery between 1992 and 2005 at the Robert-Rössle-Klinik, Charité Campus Berlin Buch (Charité-Universitätsmedizin Berlin). The mean follow-up of the patients was 115.4 months. The tumor samples were made available as tissue microarrays (TMAs) and classified by a pathologist for tumor identity, representability of the tumor, grading, Lauren-, WHO- and Ming-classification. To produce the TMAs, representative regions of the tumor samples were selected, punched into a cylinder and arranged in a paraffin block. The characterization of the cohort and the establishment of the tissue microarray has been performed and described previously [23,24,25]. This study was approved by the Institutional Review Board of the Charité (EA4/115/10).

### 2.2. S100A4 IHC

After sectioning, immunohistochemistry (IHC) staining was performed in an automated staining system (Bond III, Leica, Wetzlar, Germany) using an S100A4-specific antibody (Polyclonal Rabbit Anti-Human S100A4, Cat. #A5114, DakoCytomation, Agilent Technologies, Santa Clara, CA, USA). After pretreatment with 0.001 N citrate buffer (pH = 6.0), proteinase K and hydrogen peroxide, the primary antibody was used in a 1:200 dilution. As a secondary antibody, an anti-rabbit horse radish peroxidase (HRP) bound antibody (Anti-Rabbit IgG HRP Conjugate, Promega**^®^**, Fitchburg, MA, USA) was used in a 1:200 dilution. The staining was finalized using a 3,3-diaminobenzidine (DAB) peroxidase (HRP) substrate kit (Vector laboratories).

S100A4 expression was evaluated in blinded manner by two scientists using an immunoreactivity score (IRS) based on the percentage of stained tumor cells (0 = 0%, 1 = 1–25%, 2 = 26–50%, 3 = 51–75%, 4 = 76–100%) multiplied with the staining intensity (score 0–3 = no staining to strong staining) to give the IRS score of each sample (score 0–12). Samples with IRS > 5 were assessed as S100A4 “high” tumors, samples with <5 as S100A4 “low” tumors.

Detection and classification of patients into the molecular pathological characteristic groups (Her2Neu, status of infiltration of CD3-, CD4- and CD8-positive T-cells, PD-L1 expression (CP Score, CPS) and mismatch repair (MMR) status) detected by loss of MLH1, MSH2, MSH6, and PMS2 proteins expression, has been performed and described previously [23,24,25].

### 2.3. Cell Lines and Medium

The human AGE/S cell lines OE33 (Caucasian), MKN45 (Asian), NCI-N87 (Caucasian), OAC-P4C (Caucasian) and FLO-1 (Caucasian) were maintained in either RPMI Medium 1640 (Gibco, Thermo Fisher, Waltham, MA, USA) or DMEM Dulbecco’s Modified Eagle Medium (Gibco) supplemented with 10% fetal bovine serum (FBS) (FBS Superior, Biochrom GmbH, Berlin, Germany) in a humidified incubator at 37 °C and 5% CO2. Cells were analyzed for potential contamination and were found to be free of mycoplasm contamination. Authentication was performed by short tandem repeat genotyping at the DSMZ (German Collection of Microorganisms and Cell Cultures; Braunschweig, Germany) and was consistent with published genotypes for these cell lines.

Important characteristics of the used cell lines are summarized in Table 1.

### 2.4. Cloning and Transduction

The S100A4 mRNA was downregulated by shRNA knock-down (shRNAs, shRNA Target Gene Set, Cat # TG309668-OR, OriGene, Rockville, MD, USA). These shRNAs were stably introduced into the cells by lentiviral transduction. The lentiviral vectors were generated by transfecting the transfer vector and two helper plasmids (psPax2 and pMD2.G, Addgene, Watertown, MA, USA) into HEK293T cells (American Type Culture Collection (ATCC), Manassas, VA, USA). For this purpose, 1.5 × 10^7^ cells were seeded in T175 cell culture flasks with 30 mL growth medium (DMEM medium, Gibco + 10% FBS, Biochrom GmbH) and transduced on the following day. For the transduction solution, 2.85 mL of medium without FBS were mixed with 90 μg of polyethyleneimine (PEI, high molecular weight, anhydrous, Sigma-Aldrich, St. Louis, MO, USA) and incubated for 5 min at room temperature. Then, 30 µg of the transfer vector, 20 µg psPax2 (Addgene) and 10 µg pMD2.G (Addgene) were mixed in, incubated for 15 min at room temperature and then applied on the cells. The virus-containing cell supernatant was collected 24 and 48 h after transfection and filtered through a 0.45 µm filter (FP 30/0.45 CA-S, GE Healthcare Life Sciences, Marlborough, MA, USA). The filtrate was used directly to transduce the target cells with a multiplicity of infection (MOI) of <10. Positive cells were selected via FACS for GFP. Cells with at least 10-fold increased fluorescence intensity in the GFP channel (extinction 488 nm; emission 530/30 nm) were assessed as positive. After transduction and sorting, cells were cultured with the addition of penicillin, streptomycin and gentamicin (1:50 Pen Strep Penicillin Streptomycin, Gibco; 1:500 Gentamicin, Gibco) for one week.

### 2.5. S100A4 Expression Analysis

For RNA expression analyses, total RNA was isolated using the GeneMATRIX Universal RNA Purification Kit (Cat. No. E3598, Roboklon, Berlin, Germany) according to the manufacturer’s instructions. RNA was quantified (Nanodrop 1000 Spectrophotometer, Peqlab, Erlangen, Germany), and 50 ng of RNA was reverse transcribed with random hexamers (Invitrogen, Thermo Fisher, Waltham, MA, USA ) in a reaction mix (5 mM MgCl2 (Applied Biosystems, Thermo Fisher), 1x RT-buffer (Applied Biosystems, Thermo Fisher, Waltham, MA, USA), 250 μM pooled dNTPs (Roboklon), 1 U/μL RNAse inhibitor (Applied Biosystems, Thermo Fisher), and 2.5 U/μL Moloney Murine Leukemia Virus reverse transcriptase (Applied Biosystems, Thermo Fisher) at 23 °C for 15 min, 42 °C for 45 min, 95 °C for 5 min, with subsequent cooling to 4 °C in the TRIO thermocycler (Biometra, Göttingen, Germany). The 1:5 diluted cDNA was amplified by quantitative polymerase chain reaction (qPCR) using SYBR Green dye chemistry and the LightCycler 480 II (Roche Diagnostics, Mannheim, Germany) using the following PCR conditions: 95 °C for 2 min followed by 45 cycles of 95 °C for 7 s, 61 °C for 10 s, and 72 °C for 5 s using these primers:

GAPDH

Forward 5′-GAA GAT GGT GAT GGG ATT TC-3′

Reverse 5′-GAA GGT GAA GGT CGG AGT-3′

S100A4

Forward 5′-TGT GAT GGT GTC CAC CTT CC-3′

Reverse 5′-CCT GTT GCT GTC CAA GTT GC-3′

Data analysis was performed with the LightCycler 480 Software release 1.5.1 SP3 (Roche Diagnostics). Mean values were calculated from duplicate qRT-PCR reactions. Each mean value of the expressed gene was normalized to the respective mean of GAPDH expression.

For total protein extraction, cells were lysed with pH 7.5 RIPA buffer (50 mM Tris–HCl; pH 7, 150 mM NaCl, and 1% Igepal/NP40, 0.5% sodium deoxycholat, supplemented with cOmplete protease inhibitor cocktail tablets; Roche, Basel, Switzerland) for 5 min at room temperature and centrifuged at 12,000 rpm at 4 °C for 30 min. Protein concentration was quantified with Coomassie Plus Bradford Assay Kit (Thermo Scientific, Waltham, MA, USA), according to the manufacturer’s instructions. Lysates of equal protein concentration were mixed with 4X loading buffer NUPAGE LDS Sample Buffer (Invitrogen), boiled for 5 min at 95 °C and separated with sodium dodecyl sulfate-polyacrylamide gel electrophoresis (SDS-PAGE) (NUPAGE 10% Bis-Tris-gels, Invitrogen) and transferred to Trans Blot Turbo nitrocellulose (BIORAD, Hercules, CA, USA) membranes. Membranes were blocked for 1 h at room temperature with 5% nonfat dry milk in TBST buffer (10 mM Tris-HCl; pH 8, 0.1% Tween 20, and 150 mM NaCl). Membranes were then incubated overnight at 4 °C with S100A4 antibody (Polyclonal Rabbit Anti-Human S100A4, Cat. #A5114, DakoCytomation, Agilent Technologies, Santa Clara, CA, USA, dilution 1:1000) or β-actin antibody (Monoclonal Anti β-Actin, Mouse IgG1, A1978, Sigma-Aldrich, dilution 1:20,000), washed 3 × 10 min in TBST, followed by incubation for 1 h at room temperature with HRP-conjugated anti-rabbit IgG (Promega, Madison, WI, USA, dilution 1:10,000) or anti-mouse IgG (Pierce Antibody, Thermo Scientific, dilution 1:10,000) and 3 × 10 min washing. Antibody–protein complexes were visualized with WesternBright ECL HRP substrate (Advansta, Menlo Park, CA, USA) and subsequent imaging in a FluorChemQ developer (ProteinSimple, Bio-Techne, Minneapolis, MN, USA) and software analysis (AlphaView SA, setting Chemi white 0.5 s, aperture 16 and without illumination and aperture 0.95 for 0.3 to 3 min) or exposure to CL-Xposure Films (Thermo Fisher Scientific). Immunoblotting for β-actin served as a protein loading control.

### 2.6. Proliferation

Proliferation was analyzed in real-time using the xCELLigence system RTCA DP (Roche) mounted with 16-well-E-plates (Acela, Prague, Czech Republic). After 30 min equilibration and background measurement with 50 µL medium, 3000 FLO-1 or 50,000 NCI-N87 cells were seeded with fresh medium in each well. After letting the cells settle down for 20 min, the plate was placed in the reader and the impedance value of each well was automatically monitored 50 times every 30 min and 500 times every 15 min by the xCELLigence system for a duration of 5 days and expressed as CI (cell index) value. After normalization, the area under the curve (AUC) was calculated to compare proliferation between the different cell lines.

### 2.7. Migration

For migration analysis with FLO1 cells, the xCELLigence system RTCA DP (Roche**^®^**) mounted with 16-well-CIM-plates (CIM-Plate 16, Cat. # 5665825001, ACEA Biosciences Inc., San Diego, CA, USA) was used. A quantity of 160 µL of medium with FCS were pipetted into the lower compartment of each well and 50 µL without FCS into the upper part to allow for equilibration and normalization. Then, 50,000 FLO1 cells that had been starved on an FBS-free medium at 37 °C for 4 h were seeded with an additional 50 µL medium without FBS onto the upper insert compartment of an. Images of each insert were taken 50 times every 30 min and 300 times every 15 min.

For the cell line NCI-N87, the migration assay was carried out in transwell inserts with 8 μM pores (Corning) as described by Xie et al. [26] as the assay using CIM-plates did not work in our hands for this cell line. For equilibration, 600 μL of FBS-free medium were added per well in which the inserts were placed. Then, 300 μL of FBS-free medium were added onto the inserts and the plates were incubated for at least 30 min at room temperature. After this, the inserts were converted into a well with a 600 µL FCS-containing medium. The cells were starved for 5 h on the FCS-free medium and 30,000 cells were seeded in a 300 µL medium with 2% FCS into the insert. After 16 h, the cells were harvested for cell quantification by incubating the inserts in 600 μL of trypsin (Gibco) for 5 min and thorough rinsing of its bottom side. The FBS-containing medium from the well and the trypsin were pooled and centrifuged for 5 min at 1800 rpm at room temperature. The pellet was dissolved in 100 µL medium and placed in a 96-well plate with 25 μL CellTiter-Glo^®^ (Luminescent Cell Viability Assay, Promega) and incubated in the dark for 10 min. Samples were quantified in the Magellan infinite M200 PRO (TECAN) (integration time 1000 ms).

### 2.8. In Vitro Drug Treatment

Monotherapy was carried out on the FLO1 cell line seeded at 40,000 cells per well in a 6-well plate. After one day cells were treated with 0.5 µM, 1 µM or 2 µM niclosamide (Sigma Life Science, Burlington, MA, USA) for 24 h. Control cells were treated with the drug dissolvent (DMSO) only.

The migration of the FLO1 cell line was also examined for niclosamide which was added to the respective media in the described concentration. Analysis of migration was performed after 24 h.

### 2.9. Statistical Analysis

For data analysis and statistics GraphPad Prism 6 (GraphPad Software, San Diego, CA, USA) and SPSS Version 28 (IBM, Armonk, NY, USA) were used. The *p*-values for patient-characteristic data were determined using the Χ2 test (cross tables) or log rank test (comparison of survival curves). To calculate the significance of the experiments in cell culture experiments, an unpaired t-test was used for two groups to be compared and a one-way Anova test with Dunnett’s or Šidák’s (Figure 4F only) test for multiple comparisons. All data are expressed as mean with standard deviation.

## 3. Results

### 3.1. S100A4 Is a Negative Prognostic Marker in Caucasian Patients

Tissue material and clinical data of 277 patients (detailed clinicopathological characteristics are summarized in Table 2) were analyzed in this study (female = 99, males = 178, median age = 61.7 years). In 229 cases (82.7%) the tumor was localized in the stomach and in 48 cases (17.3%) in the esophagus or gastroesophageal junction. Patients with all tumor stages (T1 = 36, T2 = 110, T3 = 103, T4 = 28), all nodal (N0 = 72, N += 205) and all metastasis status (M0 = 199, M1 = 78) were included. Data on lymphatic infiltration was available in 258 cases, data on venous infiltration in 255 cases. Lymphatic infiltration was observed in 168 patients (60.6%), venous infiltration in 96 patients (34.6%). The mean follow-up was 115.4 months (95% CI: 104.7–126.0). The 5-year overall survival was 29.6% and the 5-year disease-specific survival was 35.3%.

After IHC staining, 74 (26.7%) tumor samples were scored with an IRS higher than 5 into the S100A4 high group and 203 (73.3%) as S100A4 low. Figure 1 shows representative pictures for samples sorted into the S100A4 high group (Figure 1 left) and in the S1004 low group (Figure 1 right).

The correlation of S100A4 expression status and patient characteristics showed significantly more S100A4 high patients in higher T-stages (*p* = 0.001), in higher N-stages (*p* = 0.012) and in M1 (*p* = 0.031) (see Table 2).

The correlation of S100A4 expression with common molecular characteristic markers such as Her2neu status, amount of tumor infiltrations lymphocytes (CD3, CD4 and CD8), MMR status and CPS showed no significant correlation (see Table 3).

Patients with S100A4 high tumors showed a mean overall survival of 35.0 months which is significantly shorter compared to S100A4 low patients with 66.0 months (*p* = 0.001) (see Figure 2A). The disease-specific survival was also reduced in S100A4 high patients (38.4 vs. 83.6; *p* < 0.001) (see Figure 2B). The impact of S100A4 as a negative prognostic factor was significant (OR 1.366 (CI 1.010–1.847; *p* = 0.043) in a multivariate Cox’s regression model using the factors: sex, UICC stage, localization, L-status, V-status and S100A4 status.

In subgroup analyses (UICC stages, nodal status, vein invasion and lymphatic vessel invasion), S100A4 was a strong predictor of reduced survival in tumors with low risk morphology: UICC I + II: S100A4 low: 96.3 months vs. 62.9 months (*p* = 0.039) (Figure 3A); N0 tumors: S100A4 low 117.2 months vs. S100A4 high 55.0 months (*p* = 0.001) (Figure 3B); L0 tumors: S100A4 low 100.6 months vs. S100A4 high 42.5 months (*p* = 0.002) (Figure 3C); V0 tumors: S100A4 low 85.4 months vs. S100A4 high 42.5 months (*p* = 0.001) (Figure 3D).

At the timepoint of initial diagnosis, 25.7% (*n* = 63) of the patients with S100A4 low-expressing tumors showed distant metastasis which are significantly less patients than in the group of S100A4 high-expressing tumors (46.9%, *n* = 15; *p* = 0.012). Regarding the patients without initial distant metastasis, 43.4% (*n* = 79) of S100A4 low patients and 76.5% (*n* = 13) of S100A4 high patients developed metachronous metastasis in the follow up period (*p* = 0.009).

### 3.2. S100A4 Induces an Invasive Phenotype

As we identified S100A4 as a strong prognostic biomarker in AGE/S in Caucasian patients, we hypothesized that S100A4 is druggable in this condition. In order to identify cell lines suitable for knock-down and inhibition of *S100A4*, we tested five different AGE/S-cell lines for their endogenous S100A4 expression (Figure 4A). Four of these cell lines were of Caucasian origin (FLO1, NCI-N87, OE33 and OAC-P4C) and one was of Asian origin (MKN45). To interconnect our in vitro studies with the formerly described patient cohort and to fill the gap in the investigation of AGE/S in the Caucasian population we decided not to choose MKN45 cells despite it showed the highest S100A4 expression both on protein and RNA levels. The OAC-P4C cell line was not used due to its difficult growth behavior in our hands and OE33 cells were excluded as S100A4 expression was very low. Therefore, we selected the two Caucasian cell lines, FLO1 and NCI-N87, which showed robust S100A4 expression to induce an shRNA knock-down for further functional assays.

Knock-down was effective in both cell lines with a reduction of mRNA-expression up to 97.3% in FLO1 (sh582, *p* = 0.0005) and up to 90.9% in NCI-N87 (sh580, *p* < 0.0001) compared to control (sh-control) (see Figure 4B,C). Since knock-down with those two mentioned constructs showed the largest reduction in S100A4 expression both on mRNA and protein level, sh580 for NCI-N87 and sh582 for FLO1 were chosen for functional assays and are indicated as shS100A4.

As measured by a live-imaging proliferation assay with the XCelligence system (Roche), knock-down showed no significant impact on proliferation in both cell lines (Figure 4D). However, compared to the control, FLO1 shS100A4 showed a 57.1% lower migration (*p* = 0.0357), and also in NCI-N87 *S100A4* knock-down decrease migration slightly (15.71% reduction compared to sh-control) but not significantly (Figure 4E).

### 3.3. Niclosamide Reduces S100A4 Expression

Next, we examined the effect of known S100A4 inhibitor niclosamide on expression and functional parameters for FLO1 cells as we saw the strongest effect of *S100A4* knock-down in this cell line. We detected a significant reduction of *S100A4* gene expression between untreated cells and cells treated with a dose of 2 µM niclosamide for sh-control cells (*p* = 0.0023) and not for shS100A4 cells (Figure 4F). Furthermore, we observed a decrease in migratory capacity for FLO1 sh-control cells treated with 1 µM niclosamide of 79.1% (*p* = 0.0430) compared to non-treated control (Figure 4G).

## 4. Discussion

With a mean 5-year survival of below 30%, the prognosis for patients with AGE/S is poor [2,3]. The poor prognosis of these tumor entities is mainly due to the frequent occurrence of distant metastases. Often, even in apparently early tumor stages, microscopic metastases are present which significantly worsen the prognosis [1,4,5]. Therefore, a strong biomarker to predict the occurrence of distant metastases is strongly needed in AGE/S. While there was already obtained promising but controversial data from Asian cohorts [10,11,12,13,14,15,16], in this study we analyzed the expression of S100A4 in AGE/S in a Caucasian cohort.

In our cohort of 277 patients, 26.7% were scored as S100A4 high expressers, which shortens overall survival (OS) and disease-specific survival (DSS) by half (OS: 66 months vs. 35 months, *p* < 0.001; DSS: 83.6 months vs. 38.4 months, *p* < 0.001). S100A4 expression was furthermore correlated with advanced tumor stages such as higher T, N and M stages. Interestingly, there was no correlation of S100A4 expression with tumor grading, Lauren classification or molecular characteristics such as MMR status, tumor lymphocyte infiltration, Combined Positive Score for PD-L1 expression in tumor cells and lymphocytes (CPS) or Her2Neu status.

The subgroup analyses showed that the negative prognostic potential of S100A4 is particularly strong in low-risk constellations. In patients that already suffer from advanced tumor stages, lymph node metastasis or lymph vessel infiltration, S100A4 expression does not impact prognosis. However, in patients with low-risk staged tumor diseases (UICC I/II, N0, L0, V0 state), S100A4 expression defines their prognosis. Especially the high rate of metachronous metastasis development in the S100A4 high group (76.5% vs. 43.4%, *p* = 0.009) highlights that analysis of S100A4 expression can help to identify high-risk patients in these cases. S100A4 high patients might benefit from extended therapy and this has to be investigated in further studies.

The use of S100A4 as a prognostic marker has been discussed extensively. There are several studies that found a link between S100A4 expression, lymph node metastasis and prognosis [10,13], but a meta-analysis of seven gastric cancer studies did not recapitulate that correlation [16]. All the studies investigated were based on Asian patient cohorts. It is known from many other studies that the tumor biology of Asian and Caucasian patients differs substantially [1,27,28,29,30]. A retrospective SEER Data (Surveillance, Epidemiology, and End Results Program) analysis of patients treated in the USA showed, that the incidence in Asian gastric cancer patients is higher than the incidence of Caucasian patients, but the survival in Asian patients is better [30]. The comparison of whole-genome sequencing and whole-exome sequencing analyzes of Asian and Caucasian patients shows that there are substantial differences in the molecular tumor characteristics such as the alteration rate of important driver genes such as APC, ARID1A, PTEN and PIK3CA [31]. For example, data from the Genomic Data Commons (GDC) database of the National Institute of Health (NIH) shows that mutations in the PI3KCA oncogene that have been associated with metastasis in several tumor entities [32,33,34] happen more frequently in Asian patients with AGE/S compared to Caucasian patients, both in absolute numbers and in comparison of frequency to other mutations [35]. A meta-analysis that included more than 2500 gastric cancer patients suggested that, for example, E-Cadherin (CDH1) polymorphism renders the Asian population more susceptible towards the development of gastric cancer [36]. This mutation can also be found in the investigated MKN45 cell line.

The most important clue to the different biology of the ethnic groups is that the response to therapy can be substantially different. The best-known example is the AVAGAST trial, analyzing the effect of bevacizumab in gastric cancer: only the non-Asian subgroup showed a benefit from bevacizumab treatment in combination with chemotherapy [37].

An evaluation of the role of S100A4 in a Caucasian population was therefore imperative necessary. Our data shows the high impact of S100A4 expression in Caucasian AGE/S. The possibility of detecting *S100A4* in gastric cancer patients’ blood plasma makes this marker also interesting as a follow-up parameter [17].

The high potential of S100A4 as a therapeutic target was shown in several studies for colorectal cancer [18,19,20]. In this study, we analyzed the effectiveness of *S100A4* inhibition by shRNA knock-down and by the S100A4 inhibitor niclosamide in AGE/S cell lines.

We screened five cell lines of AGE/S origin for their endogenous S100A4 expression. MKN45 showed the highest expression level for S100A4 which can be seen in the light of the aforementioned differences in Asian patient cohorts. For functional analysis, we used FLO1 and NCI-N87 which are cell lines of Caucasian origin to test the hypothesis that S100A4 is a mediator of metastasis. We induced a stable knock-down of *S100A4* via lentiviral transduction and found it to reduce expression by around 90% (see Figure 4). S100A4 knock-down did not change the proliferative capacity of cells which replicates findings in a study on colorectal carcinoma [38] but contrasts some others [39,40]. In migration assay, we found that *S100A4* knock-down reduced migration in the FLO1 cell line and slightly in NCI-N87 cells (see Figure 4). Migration reduction was stronger in the FLO1 cell line although baseline S100A4 expression in NCI-N87 cells exceeded that of FLO1 cells resulting in a higher fold change of expression through *S100A4* knock-down (see Figure 4). We speculate that this is due to an effect of other mediators of migration and metastasis (for instance Metastasis-associated in colon cancer 1 (MACC1) that is also expressed in some AGE/S cell lines [41] or SMAD4 that is altered in NCI-N87 cells (see Table 1)) that maintain the migratory capacity of NCI-N87 cells independently of S100A4 expression. Another explanation could be that the role of S100A4 on migration is different in the context of the tumor source, i.e.**,** FLO1 is derived from a primary tumor while NCI-N87 cells originate from a metastasis. Depending on the tumor environment, cells could be differentially primed and thus react differently when inhibiting S100A4. Future studies should consider this and systematically investigate other cell lines with different mutational backgrounds as well as tumor sources.

Next, we examined the effect of known S100A4 inhibitor niclosamide on *S100A4* expression and migratory features. We observed decreased expression of *S100A4* with treatment with niclosamide and detected reduced migration (see Figure 4). While expression reduction was not observed below a concentration of 2 µM but the effect on migration was already present at 1 µM it could be that niclosamide acts on migration through other pathways such as NFκB, mTor or Notch signaling [42,43]. Mutations in β-Catenin or other mutations driving independence from Wnt signaling are another possible explanation for the conflicting results of drug treatment.

All in all, this study can serve as a proof-of-concept to establish S100A4 as a druggable target in AGE/S. However, it is limited in regards to the transferability of the findings towards other AGE/S cell lines as we could confirm S100A4**′**s impact on migration and invasiveness only in one of the two investigated cell lines. This suggests that there are other mediators of metastasis despite S100A4 that impact the migratory phenotype of cells lines. As discussed above, it would be of great advantage to set up a systematic analysis of AGE/S cell lines from different tumor stages, origins and mutational backgrounds in order to understand the role of S100A4 in the context of other mediators of metastasis. It was also not in the scope of this study to elucidate the mechanism by which inhibition of S100A4 can reduce the migratory capacity of FLO1 cells. However, there are several possible modes of action both intra- and extracellularly that should be investigated in future studies. For example, it has been proposed that intracellular S100A4 aids with the organization of the cytoskeleton by modulating the function of non-muscular myosin-IIa filaments (NM-IIA) and thus influencing cell motility also in tumor tissues [44,45,46,47,48]. On the other hand, like other members of the S100 protein family, S100A4 can be secreted to the extracellular space where it can bind the receptor for advanced glycation end products (RAGE) and induce expression of matrix metalloproteases (MMPs) such as MMP-13 [49,50,51,52]. This can lead to increased migratory activity as has been shown in thyroid cancer [53]. Future studies on S100A4’s migratory effects in cell lines should focus on investigating these possibilities.

## 5. Conclusions

In summary, we showed for the first time in a Caucasian AGE/S population that S100A4 expression is a negative prognostic marker in these tumor entities that predicts, especially in low-risk staged tumors, the development of metachronous metastases and poor prognosis.

The in vitro phenotyping revealed the biological significance of S100A4 in two AGE/S cell lines. Therapeutic intervention in one of these cell lines with the S100A4 inhibitor niclosamide reduced cell migration. This effect has to be examined with additional cell lines and in vivo.

The high impact of S100A4 on patient survival makes it a promising target that could be an important key to improving prognosis.

## Figures and Tables

**Figure 1 cells-11-01056-f001:**
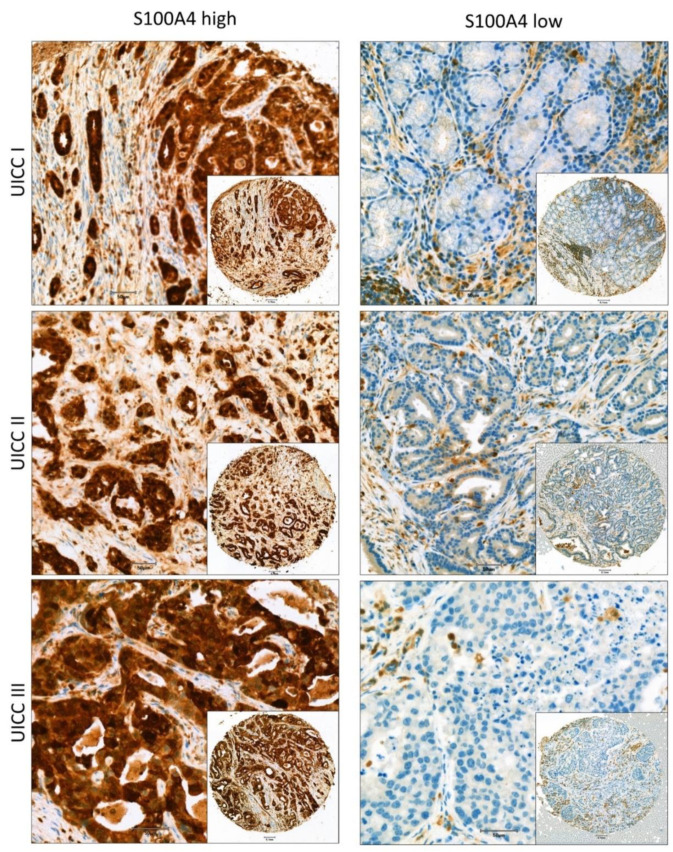
Representative pictures of IHC stained TMAs from the patient cohort from different tumor stages (UICC I, II and III) that were sorted either into the S100A4 high (**left**) or low (**right**) group. Scale bar 50 µm (inserts) and 100 µm (overview).

**Figure 2 cells-11-01056-f002:**
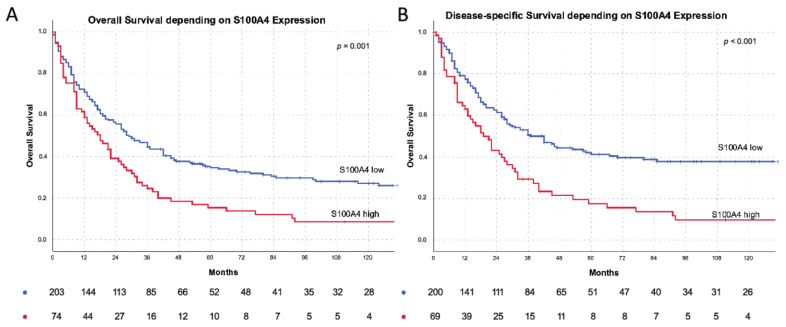
S100A4 survival and subgroup survival analysis: (**A**) S100A4 depending overall survival: *n* = 203 S100A4 low (blue): 66 months vs. *n* = 73 S100A4 high (red): 35 months mean survival (*p* = 0.001). (**B**) S100A4 depending disease specific survival: *n* = 200 S100A4 low (blue): 83.6 months vs. *n* = 69 S100A4 high (red) 38.4 months mean survival (*p* < 0.001). Tables below the Kaplan–Meyer curves describe the numbers of patients at risk.

**Figure 3 cells-11-01056-f003:**
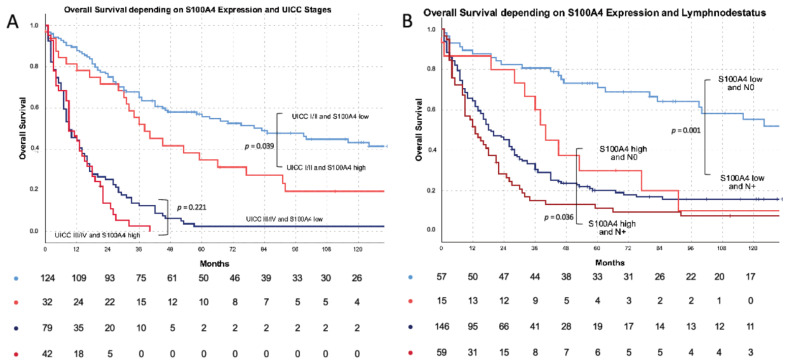
S100A4 subgroup survival analysis: (**A**) S100A4 and stage depending overall survival: in UICC I + II: *n* = 124 S100A4 low (light blue): 96.3 vs. *n* = 32 (light red): 63 months mean survival (*p* = 0.039) in UICC III+IV: *n* = 79 S100A4 low (dark blue) 19.2 vs. *n* = 41 S100A4 high (dark red): 12.7 months mean survival (*p* = 0.221). (**B**) S100A4 and lymph node status depending overall survival: in N0: *n* = 57 S100A4 low (light blue): 117.2 vs. *n* = 15 (light red): 55.0 months mean survival (*p* = 0.001) in N1: *n* = 146 S100A4 low (dark blue) 45.5 vs. *n* = 58 S100A4 high (dark red): 27.6 months mean survival (*p* = 0.036). (**C**) S100A4 and lymphatic vessel invasion status depending overall survival: in L0: *n* = 72 S100A4 low (light blue): 100.7 vs. *n* = 17 (light red): 42.4 months mean survival (*p* = 0.002) in L1: *n* = 117 S100A4 low (dark blue) 45.2 vs. *n* = 52 S100A4 high (dark red): 27.0 months mean survival (*p* = 0.097). (**D**) S100A4 and venous vessel invasion status depending overall survival: in V0: *n* = 120 S100A4 low (light blue): 85.4 vs. *n* = 38 (light red): 42.5 months mean survival (*p* = 0.001) in V1: *n* = 65 S100A4 low (dark blue) 33.7 vs. *n* = 31 S100A4 high (dark red): 17.0 months mean survival (*p* = 0.295). Tables below the Kaplan–Meyer curves describe the numbers of patients at risk.

**Figure 4 cells-11-01056-f004:**
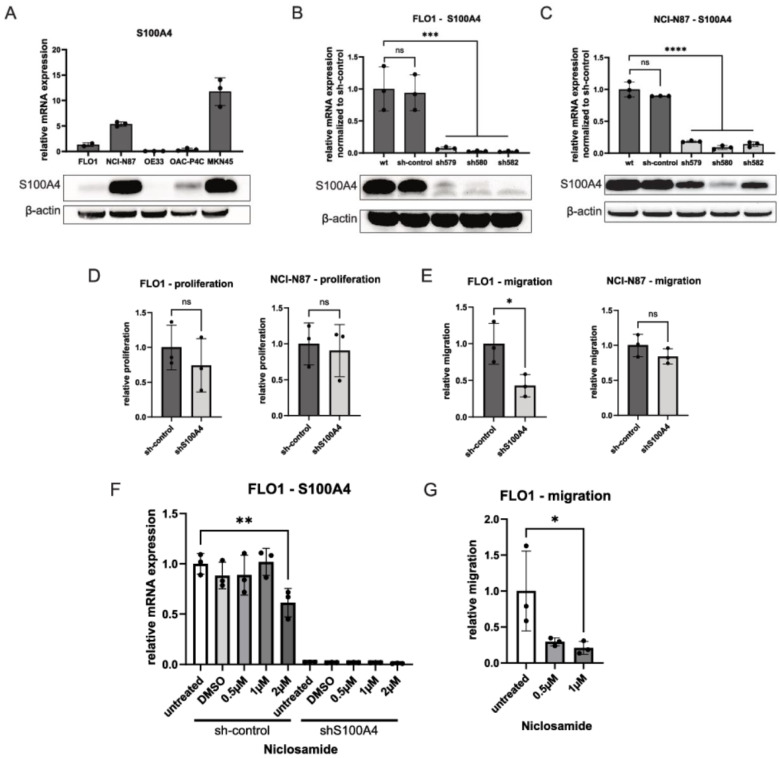
S100A4 in vitro phenotyping and functional assays. (**A**) Endogenous S100A4 expression levels (mRNA and protein) of four cell lines of Caucasian origin (FLO1, NCI-N87, OE33, OAC-P4C) and one of Asian origin (MKN45). (**B**) mRNA and protein expression levels of stable shRNA knock-down of *S100A4* in FLO1 cell line with three different constructs, a control construct (sh-control) and non-transduced cells (wt). Maximum reduction in construct sh582 (97.3%, *p* = 0.0005). (**C**) mRNA and protein expression levels of stable shRNA knock-down of *S100A4* in NCI-N87 cell line with three different constructs, a control construct (sh-control) and non-transduced cells (wt). Maximum reduction in construct sh580 (90.9%, *p* < 0.0001). (**D**) Relative proliferation of sh-control compared to the most effective knock-down line for FLO1 and NCI-N87 cell line. (**E**) Relative migration of sh-control compared to the most effective knock-down line for FLO1 and NCI-N87 cell line. Knock-down reduced migration in FLO1 cell line by 57.1% (*p* = 0.0357). (**F**) Relative S100A4 mRNA expression after treatment of FLO1 derived lines sh-control and shS100A4 with niclosamide (0.5 µM, 1 µM and 2 µM), untreated or treated with vehicle (DMS0). 2µM treatment reduced *S100A4* expression in sh-control compared to untreated control (reduction of 38.72%, *p* = 0.0023). (**G**) Sh-control cells of FLO1 cell line show reduced migration upon treatment with niclosamide (0.5 µM, 1 µM; reduction untreated vs. 1 µM: 79.1%, *p* = 0.0430), *n* = 3 for all experiments. ****: *p* < 0.0001, ***: *p* = 0.0001–0.001, **: *p* = 0.001–0.01, *: *p* = 0.01–0.05, ns: *p ≥* 0.05.

**Table 1 cells-11-01056-t001:** Characteristics of used cell lines. Information was retrieved from public sources of ATCC, DSMZ, Cellosaurus and CCLE.

Cell Line	Origin	Gender	Tumor Source	Selection of Mutant Genes
FLO1	Caucasian	male	Esophageal, primary	*BRCA2, TP53*
MKN45	Asian	female	Gastric; metastasis	*TP53, CDH1, BRCA1*
NCI-N87	Caucasian	male	Gastric; metastasis	*SMAD4, TP53, ERBB3*
OAC-P4C	Caucasian	male	Gastro-esophageal (cardial), primary	*TP53*
OE33	Caucasian	female	Barret’s (esophageal), primary	*TP53*

**Table 2 cells-11-01056-t002:** Patient characteristics of the analyzed patient cohort and distribution of S100A4 high and low tumors. Significance calculated by Χ2-Test.

Characteristic	Total		S100A4 Low		S100A4 High		*p*
	N	(%)	N	(%)	N	(%)	
Gender							
Female	99	(35.7)	72	(35.5)	27	(36.5)	0.876
Male	178	(64.3)	131	(64.5)	47	(63.5)	
Age Group							
<65 years	158	(57.0)	119	(58.6)	39	(52.7)	0.379
≥65 years	119	(43.0)	84	(41.4)	35	(47.3)	
Localization							
AEG	48	(17.3)	38	(18.7)	10	(13.5)	0.311
Stomach	229	(82.7)	165	(81.3)	48	(86.5)	
Tumor stage							
T1	36	(13.0)	36	(17.7)	0	(0.0)	<0.001
T2	110	(39.7)	80	(39,4)	30	(40.5)	
T3	103	(37.2)	70	(34.5	33	(44.6)	
T4	28	(10.1	17	(8.4)	11	(14.9)	
Node Stage							
N0	72	(26.0)	57	(28.1)	15	(20.3)	0.012
N1	85	(30.7)	70	(34.5)	15	(20.3)	
N2	55	(19.9)	34	(16.7)	21	(28.4)	
N3	65	(23.5)	42	(20.7)	23	(31.1)	
Distant Metastasis							
M0	199	(71.8)	153	(75.4)	199	(62.2)	0.031
M1	78	(28.2)	50	(24.6)	78	(37.8)	
Lymphatic vessel invasion							
L0	90	(32.5)	72	(35.5)	18	(24.3)	0.059
L1	168	(60.6)	116	(57.1)	52	(70.3)	
Unknown	19	(6.9)	NA	NA	NA	NA	
Vein invasion							
V0	159	(57.4)	120	(59.1)	39	(52.7)	0.178
V1	96	(34.7)	65	(32.0)	31	(42.9)	
Unknown	22	(7.9)	NA	NA	NA	NA	
Grading							
G1	1	(0.3)	1	(0.5)	0	(0.0)	0.06
G2	73	(25.0)	46	(22.7)	27	(36.5)	
G3	203	(73.9)	156	(156)	47	(63.5)	
Lauren Classification							
Intestinal	100	(33.9)	70	(34.5)	30	(40.5)	0.563
Diffuse	138	(52.8)	105	(51.7)	33	(44.6)	
Mixed	39	(12.5)	28	(13.8)	11	(14.9)	

**Table 3 cells-11-01056-t003:** Status of molecular characteristics markers of the analyzed patient cohort and distribution of S100A4 high and low tumors. Significance calculated by Χ2-Test.

Characteristic	Total		S100A4 Low		S100A4 High		*p*
	N	(%)	N	(%)	N	(%)	
Her2Neu							
positive	21	(7.6)	15	(7.4)	6	(8.1)	0.378
negative	212	(76.5)	152	(74.9)	60	(81.1)	
unknown	44	(15.9)	NA	NA	NA	NA	
CD3 Infiltration							
high	123	(44.4)	93	(45.8)	30	(40.5)	0.460
low	142	(51.3)	100	(49.3)	42	(56.8)	
unknown	12	(4.3)	NA	NA	NA	NA	
CD4 Infiltration							
high	103	(37.2)	74	(36.5)	29	(39.2)	0.535
low	151	(54.5)	114	(56.2)	37	(50.0)	
unknown	23	(8.3)	NA	NA	NA	NA	
CD8 Infiltration							
high	123	(44.4)	90	(44.3)	33	(44.6)	0.632
low	141	(50.9)	102	(50.2)	39	(52.7)	
unknown	13	(4.7)	NA	NA	2	NA	
Mismatch Repair System							
proficient	241	(87.0)	175	(86.2)	183	(65.8)	0.802
deficient	31	(11.2)	24	(11.8)	35	(89.7)	
unknown	5	(1.8)	NA	NA	NA	NA	
Combined Positive Score (CPS)							
<5	216	(78.0)	164	(84.1)	52	(74.3)	0.070
≥5	49	(17.7)	31	(15.9)	18	(25.7)	
unknown	12	(4.3)	NA	NA	NA	NA	

## Data Availability

Data will be made available by the corresponding author on reasonable request.

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
