# Peer review of "S100A4 Is a Strong Negative Prognostic Marker and Potential Therapeutic Target in Adenocarcinoma of the Stomach and Esophagus"

_cells, 2022, doi:10.3390/cells11061056_

Round 1

Reviewer 1 Report

In general, the manuscript shows very relevant results and is well thought out. I recommend the publication, although I have some comments.

The quality of figures 2 and 3 is very poor and should be improved. In figure 3, the p-value should appear in the figure and not in the legend, since it facilitates understanding.

In the statistical analysis section in Materials and Methods, it is specified that the one-way ANOVA test is used with the Dunnett or Šidák test (Fig. 2F only) for multiple comparisons, but there is no figure 2F.

I don't understand why the authors use different migration methods with FLO-1 and NC1-N87 cells. Is there an explanation?

The authors should specify why they use DMSO as a control in the experiments in Figure 4F. Is it because the drug is dissolved in DMSO? Add this information in Materials and Methods

In the second paragraph of the discussion (lines 374-380), add the p-values, and also in line 386.

The authors comment that S100A4 levels can be detected in plasma, but is there a correlation between plasma and primary tumor expression levels? Do plasma levels predict prognosis in the same way as primary tumor expression levels?

In Figure 4, the number of the experiments is not specified. It is important to know the “n” since in the proliferation experiments a trend is observed in the FLO-1 line although it is not significant.

The authors should hypothesize why S100A4 does not affect migration in NCI-N87 and does in FLO-1, and even more so when the baseline expression of NCI-N87 is much higher, so the fold-change expression between baseline and the knockout is greater. What would happen to the OAP-PC4C cells? Do the authors expect the same features for OAP-PC4C cells as FLO-1 cells or as NCI-N87 cells? Could the authors speculate which patients might benefit from niclosamide treatment?

What is the mechanism by which the S100A4 affects migration?

Reviewer 2 Report

In this manuscript, the authors showed the prognostic value of S100A4 gene expression in Caucasian AGE/S cancer patients. The preliminary in-vitro experiments showed the oncogenic behavior of S100A4 and the potential of targeting it. The study is interesting with clinical worth. Please address the following points to fill up the small gaps in the study.

  1. Though the histology is performed in the samples from the German hospital, are the authors sure about the ethnicity of all the 277 patients?
  2. Line 269-271: is this observation consistent with the AGE/S Asian patient cohorts and other cancer types, i.e., differential expression of S100A4 in different staging?

  3. Please show the representative examples of S100A4 IHC staining for different TNM staging as well.

  4. Scale bar missing in Figure 1.

  5. For Figures 2 and 3, please increase the font size of the name of the groups in KM plots. Making a table as well for the survival information in Figure 3 will be helpful for the readers.

  6. Four cell lines of Caucasian origin (FLO1, NCI-N87, OE33, and OAC-P4C) and one of Asian origin (MKN45) are mentioned in the paper. What is the difference among these cell lines regarding common mutations in AGE/S cancer, e.g., TP53, CDH1, ARID1A, PIK3CA, KRAS...(authors may use CCLE). This information may help us understand the varying levels of S100A4 in these cell lines. Similarly, is there a difference in the mutation status of these genes between Asian and Caucasian patients? Authors can use sequencing data from different cancer datasets worldwide to find this information. 

  7. Shedding more light in the discussion about the similarities and differences between Asian and caucasian AGE/S cancer patients will be insightful.

  8. Limitations and future directions for this study should be mentioned.

Round 2

Reviewer 1 Report

The authors have covered all my questions/suggestions. I only suggest improving the quality of Figure 4 as well.